# RDVI: A Retrieval–Detection Framework for Verbal Irony Detection

**Zhiyuan Wen** [1,2], **Rui Wang** [1,2], **Shiwei Chen** [1,2], **Qianlong Wang** [1,2], **Keyang Ding** [1,2], **Bin Liang** [1,2]
**and Ruifeng Xu** [1,2,3,*]

[1] Harbin Institute of Technology, Shenzhen 518000, China; wenzhiyuan@stu.hit.edu.cn (Z.W.);
ruiwangnlp@outlook.com (R.W.); chenshw@pcl.ac.can (S.C.); 21B351012@stu.hit.edu.cn (Q.W.);
keyang.ding@stu.hit.edu.cn (K.D.); bin.liang@stu.hit.edu.cn (B.L.)
[2] Guangdong Provincial Key Laboratory of Novel Security Intelligence Technologies, Shenzhen 518000, China
[3] Peng Cheng Laboratory, Shenzhen 518000, China
[*] Correspondence: xuruifeng@hit.edu.cn

**Abstract:** Verbal irony is a common form of expression used in daily communication, where the intended meaning is often opposite to the literal meaning. Accurately recognizing verbal irony is essential for any NLP application for which the understanding of the true user intentions is key to performing the underlying tasks. While existing research has made progress in this area, verbal irony often involves connotative knowledge that cannot be directly inferred from the text or its context, which limits the detection model's ability to recognize and comprehend verbal irony. To address this issue, we propose a Retrieval–Detection method for Verbal Irony (RDVI). This approach improves the detection model's ability to recognize and comprehend verbal irony by retrieving the connotative knowledge from the open domain and incorporating it into the model using prompt learning. The experimental results demonstrate that our proposed method outperforms state-of-the-art models.

**Keywords:** verbal irony detection; connotative knowledge; prompt learning; deep learning; natural language processing





## 1. Introduction

*"Irony is a device of both mind and language for acknowledging the gap between what is expected and what is observed"* [1]. The scholarly investigation of irony has an ancient history and an extensive foundation. However, modern studies on irony are mainly focused on its relationship with thought and language [2]. Irony encompasses several distinct concepts: *Socratic irony*, *situational irony* (extended as the *irony of fate* and *dramatic irony*), and *verbal irony* [3,4]. *Verbal irony* is widely used in everyday communication, especially on social media platforms. It is often described as an utterance (a textual expression or linguistic expression) that expresses the polar opposite of what it really means. While it is common for many researchers to use irony or sarcasm to refer to verbal irony, and it is acceptable to treat sarcasm interchangeably with irony when discussing it [5], it is essential to note that these are similar but distinct. This paper uses the term "verbal irony" uniformly to ensure accurate understanding. Accurate and automated identification of verbal irony may allow users' genuine intentions to be understood, thereby facilitating numerous tasks in natural language processing, including e.g. sentiment analysis [6], hate speech detection [7], and argument detection [8].

Verbal irony is a highly nuanced and intricate rhetorical device. Expressions of verbal irony frequently encompass connotative knowledge, which includes commonly accepted conceptual knowledge such as common sense, as well as knowledge that is specific to certain groups and subject to modification over time [9]. To better illustrate the connotative knowledge in verbal irony, we show a specific example in Table 1. Merely examining the first sentence in isolation does not provide sufficient evidence to ascertain whether this is

an instance of verbal irony. Even when viewed in conjunction with its context information, it may still be challenging to detect. However, when one is aware of the connotative knowledge that Samsung mobile phones have been known to spontaneously combust due to battery problems (https://en.wikipedia.org/wiki/Samsung_Galaxy_Note_7 accessed on 5 May 2023), the ironic intent becomes evident.

**Table 1.** An example of the connotative knowledge in verbal irony.

| Category | Content |
|---|---|
| Verbal Ironic Expression | The terrorist's weapons and ammunition have arrived. |
| Context Information | Samsung released the first mass-produced folding screen mobile phone in history |
| Connotative Knowledge | Samsung note 7 mobile phone battery faults. |

The presence of connotative knowledge poses a challenge to the accurate detection of complex verbal irony expressions by the model. Although verbal irony expressions may be detected through other features, such as inconsistency, the absence of this knowledge makes it difficult for the model to fully comprehend the user's genuine intentions. The performance of detecting verbal irony has been significantly improved with the introduction of deep learning [10,11], specifically the development of pre-trained models [12,13]. However, current model techniques are limited in their ability to identify and acquire connotative knowledge.

Several researchers have noticed the significance of connotative knowledge in detecting verbal irony. Still, their approach involves equating this knowledge with common sense and incorporating a lexico-semantic knowledge base [9] or knowledge generator [14] into the model. However, such attempts do not effectively address the issue of the model's lack of connotative knowledge. On the one hand, connotative knowledge is often implicit in the expression. It is not always readily available in a knowledge base. For example, the connotative knowledge in the above example in the Table 1 is challenging to obtain directly from the knowledge base automatically. On the other hand, connotative knowledge is not always static, and much of it is closely linked to internet memes [15], which can evolve over time and impact the detection of verbal irony. For instance, the smiley emoji, initially intended to convey happiness or positivity, has acquired a mocking connotation in some contexts. Other forms of connotative knowledge, such as those related to the COVID-19 pandemic (https://en.wikipedia.org/wiki/COVID-19_pandemic accessed on 5 May 2023) or the Russian–Ukrainian war (https://en.wikipedia.org/wiki/Russo-Ukrainian_War accessed on 5 May 2023), emerge in response to specific events. To alleviate this dilemma, we draw upon research on open-domain question answering tasks to inspire the approach to identify and retrieve connotative knowledge in verbal ironic expressions.

*Open-domain question answering* (OpenQA) is a task that aims to answer a given question without any specific context provided [16]. The existing OpenQA system usually consists of two primary components: *Retriever* and *Reader* [17,18]. OpenQA typically operates on unstructured text and is not limited to a particular domain. Generally, a question answering system starts by retrieving relevant documents from open domains to serve as context.

Inspired by this, we reformulate verbal irony detection as an open-domain question answering task, where the retrieval of connotative knowledge corresponds to the Retriever component, and verbal irony detection based on relevant connotative knowledge corresponds to the Reader component. In this paper, we propose a Retrieval–Detection framework for Verbal Irony, called *RDVI*, which is a Retrieval–Detection system that employs connotative knowledge to improve the model's capacity to detect verbal irony. The framework is composed of two stages.

In the first stage, we aim to identify documents that contain connotative knowledge that is relevant to the given text. To achieve this, we retrieve documents and select the k most similar segments based on their semantic similarity to the text and its context. These segments serve as potential sources of connotative knowledge. In the second stage, we leverage connotative knowledge via prompt learning to improve the model's ability to comprehend text semantics, thereby enhancing its capacity to detect verbal irony.

The main contributions of our work can be summarized as follows:

- We propose a Retrieval–Detection framework that leverages connotative knowledge to enhance the model's ability to recognize and comprehend verbal irony.
- We utilize prompt learning to explicitly incorporate connotative knowledge into the model, thereby enhancing the model's capacity to comprehend text semantics.
- Our approach is compared to several baseline methods, and the quantitative and qualitative results demonstrate that it achieves state-of-the-art performance in detecting verbal irony.

The remaining parts of this paper are organized as follows. In Section 2, we review some related works on verbal irony detection and OpenQA to facilitate comprehension. We elaborate on our proposed framework, *RDVI* in Section 3. We describe our experiments in Section 4. Finally, we conclude our work in Section 5.

## 2. Related Work

### 2.1. Verbal Irony Detection

Accurately recognizing verbal irony is critical to understanding people's true intentions. Researchers have become increasingly interested in automating the detection of verbal irony with the development of machine learning. Numerous datasets have been created to aid in the study of verbal irony detection. Some of these datasets rely on specific tags (such as hashtags) [19–21] or particular social media accounts [22] to collect data automatically. While this method can quickly generate a large-scale dataset, the quality is difficult to ensure. An alternative method is to collect texts from platforms such as Twitter (https://twitter.com/ accessed on 5 May 2023) [23,24], Amazon (https://www.amazon.com/ accessed on 5 May 2023) [25], and Guanchazhe (https://www.guancha.cn/ accessed on 5 May 2023) [26] and deliver them to human annotators for labeling. While this manual approach may provide high-quality datasets, the quantity of data is limited. Currently, mainstream research is primarily focused on binary-category-based verbal irony tasks.

Early research on detecting verbal irony relied on rule-based approaches. Some scholars used smiley emoticons [27], and some verbal or gestural indicators such as heavy punctuation and quotation marks [28] to identify verbal irony. Meanwhile, other researchers treated hashtags in tweets as a vital signal of verbal irony [29], while some regarded positive sentences containing negative phrases as verbal irony utterances [30]. While these methods may yield satisfactory outcomes when applied to specific texts or scenarios, they are prone to errors and cannot be extended to other situations.

In the subsequent studies, the researchers employed various manual features to identify verbal irony, including lexical factors [9,31], semantic factors [32], and statistical factors [22], in combination with traditional machine learning techniques such as support vector machines (SVM) [33], decision trees [34], and logistic regression (LogR) [35]. However, the traditional machine learning approach has limitations because it relies on complex feature engineering, which is time consuming and requires significant knowledge and expertise.

Several works based on deep learning have been developed in recent years. For example, Amir et al. [36] proposed the *CUE-CNN* model, which utilizes a convolutional neural network to consider the speaker's identity and the content of the message. In another work, Ghosh et al. [10] investigated linguistic and psychological contexts using a CNN + Bi–LSTM neural network model.

Tay et al. [37] proposed the *MIARN* model, which employs multi-dimensional intra-attention to capture incongruity information between sentences. Similarly, Xiong et al. [11]

used a self-matching attention-based model to examine word-to-word interactions, followed by a low-rank bilinear pooling to concatenate congruity with sentence composition information and reduce redundancy.

González et al. [12] utilized the Transformer [38] Encoder to contextualize pre-trained Twitter word embeddings to detect verbal irony. Babanejad et al. [39] modified BERT's architecture and retrained it with affective and contextual features. The resulting model, Adversarial and Auxiliary Features-Aware BERT (AAFAB), is a unified framework that employs adversarial training and BERT to generate meaningful sentence representations [40]. By using users' historical tweets and conversational neighborhoods, Joan Plepi et al., constructed a heterogeneous social network. They subsequently introduced a graph attention-based model to examine the importance of interaction and contextual information in detecting verbal irony [41]. Additionally, Wen et al. [42] incorporated sememe knowledge and auxiliary information to improve the BERT model's performance when detecting verbal irony. Savini et al. [13] explored a transfer learning framework that enhances the effectiveness of the BERT model by fine-tuning it on intermediate tasks that are rich in data, such as emotion detection and sentiment classification. Wang et al. [43] investigate verbal irony detection from an unsupervised perspective. They explore a masking and generation paradigm within the context to extract contextual incongruities that contribute to learning verbal ironic expressions.

### 2.2. Open-Domain Question Answering

In traditional *Open-domain question answering* (*OpenQA*) systems, a pipeline consisting of three stages is typically employed: *Question Analysis*, *Document Retrieval*, and *Answer Extraction* [44,45]. The Question Analysis step of an OpenQA system takes a natural language input question and attempts to reformulate it to provide search queries for a later Document Retrieval. Moreover, Question Analysis organizes the query into categories to determine the type(s) of the anticipated answer, which directs the Answer Extraction step. In the Document Retrieval step, using the generated search queries, the system searches relevant documents or passages. Both general information retrieval methods such as TF-IDF [46] and BM25 [47], and methods created especially for online search engines such as Google (www.google.com accessed on 5 May 2023) and Bing (www.bing.com accessed on 5 May 2023), are often used. Finally, during the Answer Extraction stage, the system extracts the final answer from the pertinent documents acquired in the previous step.

With the development of deep learning and Machine Reading Comprehension (MRC) technology [48], the OpenQA system has evolved into a "*Retriever-Reader*" architecture [17,18]. The Retriever component mainly focuses on retrieving relevant documents based on a given question, akin to an information retrieval system. The Reader component primarily employs reading comprehension technology to extract the final answer from the retrieved documents.

Contemporary approaches to the Retriever can be broadly categorized into three types: *Sparse Retriever, Dense Retriever, and Iterative Retriever*. *Sparse Retriever* mainly relies on classical information retrieval techniques to retrieve documents [17,49,50]. In contrast, *Dense Retriever* employs deep learning models to learn dense semantic representations of documents, which are then used to retrieve relevant documents [51,52]. *Iterative Retriever* searches for relevant documents in multiple steps [53,54]. *Readers* can be classified into two types: *Extractive Readers* and *Generative Readers*. *Extractive Readers* predict answer spans from the retrieved documents [51], while *Generative Readers* use sequence-to-sequence (Seq2Seq) models to generate answers in natural language [55]. To improve the accuracy of OpenQA systems, additional auxiliary modules such as *Document Post-processing* and *Answer Post-processing* can be integrated. *Document Post-processing* can refine and re-rank retrieved documents [18,56], while *Answer Post-processing* can select the best answer from multiple options [57,58].

## 3. Approach

First, we briefly formalize the problem of verbal irony detection as follows. Given a text $x^c$, where $x^c = (x_1^c, x_2^c, \ldots, x_n^c)$, $x_i^c$ is the *i*-th word in the text, where $i \in [1, N]$, and $N$ is the length, the goal is to predict the verbal ironic label $y \in \{0, 1\}$ corresponding to $x^c$. Previous research has demonstrated the importance of context information in modeling the semantic context and background knowledge of a given text. In this paper, the context information is defined as a text sequence $x^t = (x_1^t, x_2^t, \ldots, x_m^t)$ without loss of generality, and $M$ is the length of the context information.

The overall architecture of our proposed framework is shown in Figure 1. In the retrieval stage, our first step is to identify relevant documents containing connotative knowledge through the retrieval and then find the *K* sentences most similar to a given text and its context by computing semantic similarity. These sentences serve as anchors, around which we sample adjacent context sentences to form text fragments. We consider these k fragments as candidate knowledge. In the detection stage, we utilize the prompt learning framework to enhance the pre-trained language model's (PLM) ability to model text semantics, ultimately leading to improved verbal irony detection.

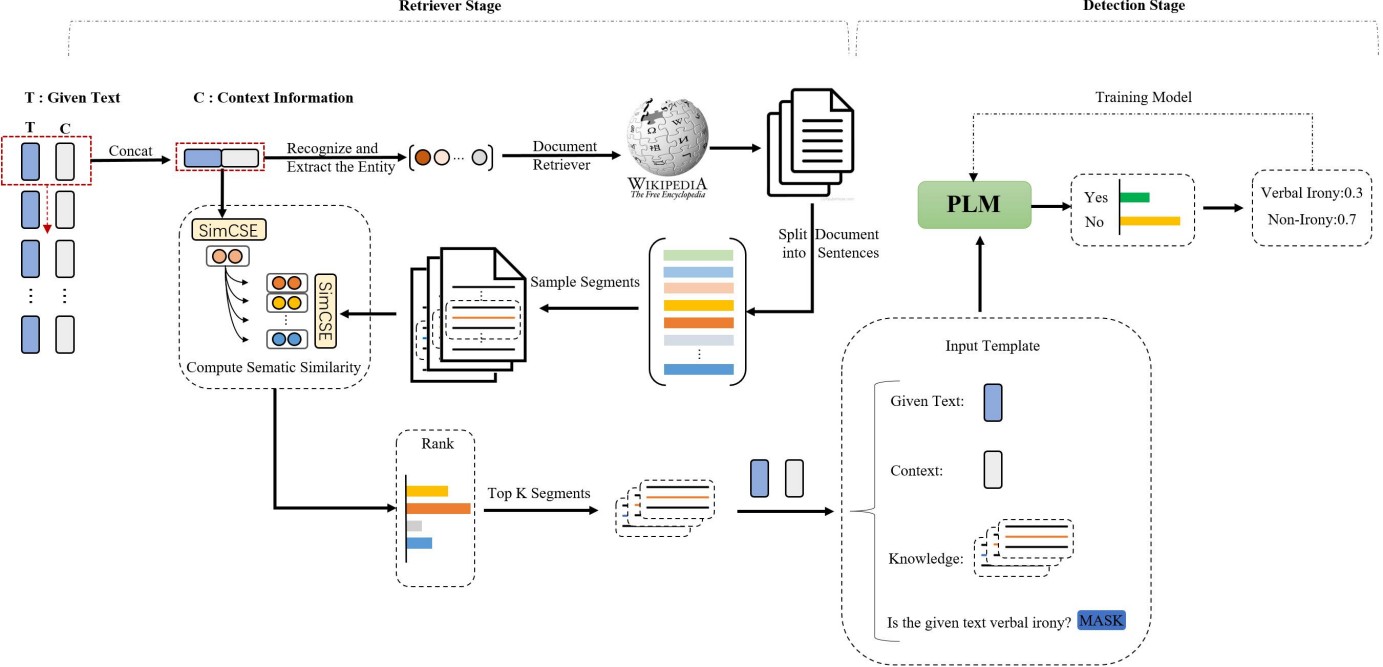

**Figure 1.** The architecture of RDVI framework.

### 3.1. Retrieval Stage

We follow the OpenQA approach to extract connotative knowledge from the given text. However, the verbal irony detection task poses a greater challenge than a typical question–answer task, as the text is not structured as a question. This makes it harder to identify connotative knowledge. To reduce unnecessary errors, instead of transforming the text into questions, we directly extract entity information since knowledge is typically associated with entities. Therefore, the first step is to extract the entities involved in the given text and its context:

$$E = \text{Entiy\_Recognition}([x^c; x^t]) \qquad (1)$$

where Entiy_Recognition is a tool used to recognize and extract entities in sentences.

These entities $E$ are then employed to retrieve associated documents from the open domain. In this paper, we use Wikipedia as the source of the documents:

$$D = \text{Retrieve}(E) \tag{2}$$

where Retrieve is a tool used to retrieve the related document for the entity.

In instances where no entity can be extracted from the given text and its context, we rely on the text and its context to retrieve pertinent documents:

$$D = \text{Retrieve}([x^c; x^t]) \tag{3}$$

Next, we preprocess the retrieved documents, retaining only the textual information. To eliminate redundant information from the document and enable the model to concentrate more on pertinent knowledge, we split the document into segments. To ensure that the segments are more coherent, we first divide the document into sentences:

$$S = \text{Split}(D) \tag{4}$$

where Split is a function to split the document into sentences.

After shuffling, we leverage each sentence as an anchor and employ a window of size 3 to sample segments. The preceding and following sentences adjacent to these anchor points are sampled:

$$s_j^p, s_j^n = \text{Sample}(s_j) \tag{5}$$

where $s_j^p$ is the previous sentence and $s_j^n$ is the next sentence adjacent to $s_j$, and each:

$$seg_j = [s_j^p; s_j; s_j^n] \tag{6}$$

To achieve greater precision in identifying pertinent knowledge, we initially measure the semantic similarity between a segment and the given text and its context. In this study, we utilize the SimCSE [59] model to compute semantic similarity. SimCSE utilizes dropout as a data augmentation technique to generate positive pairs and applies contrastive learning to improve sentence representation learning:

$$\ell_{SimCSE} = -\sum_{i=1}^{N} \log \frac{e^{\text{sim}(h_i^* \cdot h_i^+ / \tau)}}{\sum_{j=1}^{N} e^{\text{sim}(h_i^* \cdot h_j^+ / \tau)}} \tag{7}$$

where $h_i$ refers to the encoder representation of input $X_i$, while $h_i^*$ and $h_i^+$ represent the hidden states produced by dropout. The function sim corresponds to the cosine similarity between vectors $u$ and $v$, calculated as $\text{sim}(u, v) = \frac{u^T \cdot v}{\|u\| \cdot \|v\|}$, and $\tau$ is a temperature hyperparameter.

By performing a calculation, the top K textual segments composed of three sentences are used as a candidate for connotative knowledge:

$$c_j = \text{sim}(M_{SimCSE}(s_j), M_{SimCSE}(X_i)) \tag{8}$$

where $M_{SimCSE}$ is the trained model SimCSE.

$$S_i^{top} = \text{Find\_Top}(Seg_i, C_i) \tag{9}$$

The details of the retrieval stage are shown in Algorithm 1.

### 3.2. Detection Stage

We employ prompt learning, based on the *Openprompt* [60] framework, to identify verbal irony. This approach can bridge the gap between fine-tuning and pre-training processes, thereby facilitating the model's ability to model ironic expressions.

---

**Algorithm 1:** Recognize and retrieve relevant connotative knowledge

---

**Input:** Input corpus text $X^C$ and context information $X^T$, where
$X^c \in X^C, X^c = (x_1^c, x_2^c, \dots, x_n^c)$, and its context information
$X^t \in X^T, X^t = (x_1^t, x_2^t, \dots, x_m^t)$

**Output:** The relevant segment corpus containing connotative knowledge $P$

1 Create $Seg \leftarrow \varnothing$;
2 **for** $i \leftarrow 1\ to\ |X^C|$ **do**
3   Concatenate $X_i^c$ and $X_i^t$: $X_i = Cat(X_i^c, X_i^t)$;
4   Recognize and extract the entity in $X_i$: $E_i = \text{Entiy\_Recognition}(X_i)$ (Equation (1)) ;
5   Create $D_i \leftarrow \varnothing$;
6   **if** $E_i \neq \varnothing$ **then**
7     **for** $j \leftarrow 1\ to\ |E_i|$ **do**
8       Retrieve relevant documents $D^e = \text{Retrieve}(E_i^j)$ (Equation (2)) ;
9       **if** $D^e \neq \varnothing$ **then**
10        Add $D^e$ into $D_i$: $D_i \leftarrow D^e$;
11      **end**
12    **end**
13   **else**
14    Use $X_i$ to retrieve relevant documents $D^x = \text{Retrieve}(X_i)$ (Equation (3)) ;
15    Add $D^x$ into $D_i$: $D_i \leftarrow D^x$;
16   **end**
17   Create $S_i \leftarrow \varnothing$;
18   **for** $z \leftarrow 1\ to\ |D_i|$ **do**
19     Split document into sentences: $s^z = \text{Split}(D_i^z)$ (Equation (4)) ;
20     Add $s^z$ into $S_i$: $S_i \leftarrow s^z$;
21   **end**
22   Shuffle $S_i$;
23   Create $Seg_i \leftarrow \varnothing$;
24   **for** $q \leftarrow 1\ to\ |S_i|$ **do**
25     Set $S_i^q$ as anchor: $anc^q \leftarrow S_i^q$;
26     Take $anc^q$ as the center, sampling the textual segment:
       $seg^q = \text{Sample\_Segment}(anc^q)$ (Equation (5));
27     Add $seg^q$ into $Seg_i$: $Seg_i \leftarrow seg^q$;
28   **end**
29   Create $C_i \leftarrow \varnothing$;
30   **for** $p \leftarrow 1\ to\ |Seg_i|$ **do**
31     Calculate semantic similarity $c^p = \text{sim}(M_{SimCSE}(Seg_i^p), M_{SimCSE}(X_i))$ (Equation (8));
32     Add $c^p$ into $C_i$: $C_i \leftarrow c^p$;
33   **end**
34   Find the top K relevant segments: $S_i^{top} = \text{Find\_Top}(Seg_i, C_i)$ (Equation (9));
35   Add $S_i^{top}$ into $Seg$: $Seg \leftarrow S_i^{top}$;
36 **end**

---

To begin with, we create a template that transforms the input text into a prompt. This template consists of a textual string that includes a prompt description and several slots. The template's format is illustrated below:

- Given Text: $\boxed{\text{Text}}$
- Context: $\boxed{\text{Context}}$
- Knowledge: $\boxed{\text{Connotative Knowledge}}$
- Is the given text verbal irony? $\boxed{\text{MASK}}$

The template includes four slots that must be filled with the appropriate information. The first slot, marked in green ($\boxed{\text{Text}}$), should contain the given text $x^c$. The second slot, marked in yellow ($\boxed{\text{Context}}$), should be filled with the context information $x^t$. The third slot, marked in red ($\boxed{\text{Connotative Knowledge}}$), should contain the related segments *Seg*. The fourth slot, marked in blue ($\boxed{\text{MASK}}$), represents the location of the masked token, which needs the model for prediction. We leverage a function to generate the input $\hat{x}$:

$$\hat{x} = f_{prompt}(x^c, x^t, seg). \tag{10}$$

Subsequently, through a pre-trained language model $M$, we encode the input $\hat{x}$ and calculate the probability distribution over the entire vocabulary for *MASK* token, then maximize the probability score:

$$h^{mask} = M(\hat{x}; \Theta) \tag{11}$$

$$p(\hat{y}) = p_\Theta(MASK|\hat{x}) = softmax(W_\Theta h^{mask}) \tag{12}$$

where $\Theta$ is the parameter of model $M$, the $\hat{y} \in \hat{Y}$ where $\hat{Y}$ is a subset of the words in the vocabulary of $M$. To establish a connection between words and their respective class labels, we design a *verbalizer* as an injective function $\hat{Y} \rightarrow Y$.

Finally, we calculate the predicted label probability through the softmax function and leverage the cross entropy as the loss function in the optimization of our model:

$$L = \frac{1}{N}\sum_i L_i = -\frac{1}{N}\sum_i \hat{y} log(p(\hat{y})) \tag{13}$$

## 4. Experiments and Analysis

In this section, we evaluate the effectiveness and efficiency of our proposed model using the benchmark dataset *GuanSarcasm* (Guanchazhe Chinese Sarcasm Dataset) [26] for verbal irony detection and then report the empirical results.

### 4.1. Dataset

*GuanSarcasm* was manually annotated by five annotators using a majority voting strategy and obtained from the news and opinion website *Guanchazhe* (https://www.guancha.cn/ accessed on 5 May 2023). This site reports on current events, particularly political and international stories that often elicit heated debates, making it an ideal source for researching verbal irony detection. GuanSarcasm contains 4972 comments from 720 news articles. We split the dataset into training and testing sets to avoid the problems of K-fold cross-validation, which is prone to high variability that can lead to suboptimal model selection decisions and unpredictable behavior in the estimated prediction error. The details of the corpus are presented in Table 2. We assessed the overall performance of our model by measuring the accuracy and F1 score, where the F1 score is defined as $2(p \cdot r)/(p + r)$, with $p$ and $r$ representing precision and recall, respectively.

**Table 2.** Corpus statistics and verbal irony distribution for the new division of *GuanSarcasm* dataset.

|  | Category | Comment | News | Comment (AVG) | Title (AVG) |
|---|---|---|---|---|---|
| **Train** | **Verbal Irony** | 2222 | 640 | 23.966 | 24.251 |
|  | **Non-Irony** | 2222 | 637 | 22.383 | 24.259 |
| **Test** | **Verbal Irony** | 264 | 80 | 23.098 | 25.001 |
|  | **Non-Irony** | 264 | 80 | 29.220 | 24.996 |

### 4.2. Settings and Baseline

In this study, we utilized *TexSmart* (https://ai.tencent.com/ailab/nlp/texsmart accessed on 5 May 2023) to extract entities and applied fuzzy matching to search for relevant documents. We trained *SimCSE* using 2 million Wikipedia sentences. The maximum sequence length was set to 512, and the model was fine-tuned for two epochs with a batch size of 32. We used the *Adam* optimizer with a learning rate of $3 \times e^{-5}$ and trained the model on a single V100 GPU. The basic encoder used was $BERT_{BASE}$, and the maximum sequence length was the same as *SimCSE*. The model was fine-tuned for 20 epochs with a batch size of 16 on a single V100 GPU, using an *Adam* optimizer with a learning rate of $2 \times e^5$. To ensure the stability of our model, we ran it five times with different random seeds and took the average result as the final result.

We compared our model with several verbal irony detection methods to assess its effectiveness. These methods include:

- *CNN–LSTM–DNN*[61], which is a combination of CNN, LSTM, and a fully connected DNN layer for semantic modeling.
- *MIARN* and *SIARN* [37], which use a multi-dimensional intra-attention objective and a single-dimensional intra-attention objective, respectively, in a recurrent network to detect contrastive sentiment, situations, and incongruity based on intra-sentence similarity.
- *SMSD* and *SMSD–BiLSTM* [11], where *SMSD* is a self-matching network that captures incongruity information and compositional information of sentences based on a modified co-attention mechanism, and *SMSD-BiLSTM* employs a bi-directional *LSTM* to capture compositional information for each input sentence.
- *BERT* [62], which is a widely used pre-trained language model based on the Transformer architecture [38] and has achieved impressive performance in many NLP tasks.
- $BERT^{SSAS}$ [42], which incorporates sememe knowledge and auxiliary information into BERT to construct the representation of text.
- *ChatGPT* is a large language model trained by OpenAI. It has a good in-context learning (ICL) [63] ability. We select two samples for each category and use the API of OpenAI https://openai.com/ (accessed on 5 May 2023) for testing.
- *ChatGPT + Retrieval* is a method that replaces the detection component of our proposed method with ChatGPT.
- v [47], to further analyze our method, we also attempted to replace *SimCSE* with BM25 to compute semantic similarity.

### 4.3. Experimental Results

Table 3 presents an overview of the experimental results. Our proposed model $RDVI_{SimCSE}$ achieves the best performance on both datasets, with an F1 score that outperforms the previous best approach, $BERT^{SSAS}$, by *3.48%*, and an accuracy improvement of *3.59%*. These results suggest that our model effectively retrieves relevant segments as connotative knowledge to improve the model's semantic comprehension and enhance its ability to detect verbal irony. The method $RDVI_{SimCSE}$ that employs *SimCSE* to calculate semantic similarity achieves a better detection performance than the method $RDVI_{BM25}$ using *BM25*. It proves that more relevant text fragments can be found as connotative knowledge through *SimCSE*. To assess the efficacy of our model's performance, we utilized

the parameter configuration outlined in the original $BERT^{SSAS}$ method (considered the best baseline). We then conducted training using 25 distinct random seeds to produce a range of results. Subsequently, we performed a two-tailed t-test to compare the F1 scores of our model with those of $BERT^{SSAS}$, assessing the statistical significance of the differences. The result indicates that our method $RDVI_{SimCSE}$ is statistically significant at the 0.001 level ($t = 11.353$, $p = 3.267 \times e^{-6} < 0.001$) for the best baseline $BERT^{SSAS}$. It is worth noting that the retrieval component is also evidently effective for the large language model.

**Table 3.** Experimental results on *GuanSarcasm* dataset.

| Approaches | Precision | Recall | F1 Score | Accuracy |
|---|---|---|---|---|
| CNN-LSTM-DNN | 65.29% | 65.28% | 65.27% | 65.28% |
| MIARN | 68.12% | 67.92% | 67.84% | 68.50% |
| SIARN | 70.39% | 70.34% | 70.32% | 70.34% |
| SMSD | 68.51% | 68.51% | 68.50% | 68.50% |
| SMSD-BiLSTM | 71.13% | 70.96% | 70.91% | 70.96% |
| BERT | 75.21% | 76.39% | 75.68% | 75.57% |
| $BERT^{SSAS}$ | **78.79%** | 74.55% | 75.93% | 75.95% |
| ChatGPT | 62.60% | 75.93% | 71.11% | 71.32% |
| ChatGPT + Retrieval | 64.12% | 84.00% | 75.58% | 75.91% |
| $RDVI_{BM25}$ | 75.57% | 81.15% | 78.95% | 78.97% |
| $RDVI_{SimCSE}$ | 71.37% | **85.39%** | **79.41%** | **79.54%** |

To analyze the contribution of the essential components of our proposed model, we conducted an ablation experiment on our model. As shown in Table 4, when removing the *Retrieval* component, the model's performance degrades the most. The result has shown that retrieval can enhance the model's understanding of ironic semantics, leading to an improvement in its performance. Solely using entities from a given text or its context may result in a degradation of the model's performance, indicating that connotative knowledge may exist in a given text or its context. Retrieval alone may not suffice to restore the expression semantics fully. Moreover, the research revealed that prompt learning brings only limited improvement to the model, possibly due to the small size of our model, which hinders the effective utilization of prompt learning.

**Table 4.** Ablation experiments.

| Approaches | Precision | Recall | F1 Score | Accuracy |
|---|---|---|---|---|
| $RDVI_{SimCSE}$ w/o $\ell_{Retrieval}$ | 67.18% | 83.41% | 76.56% | 76.86% |
| $RDVI_{SimCSE}$ w/o $\ell_{Prompt}$ | **75.19%** | 82.08% | 79.32% | 79.35% |
| $RDVI_{SimCSE}$ w/o $\ell_{E_{Context}}$ | 74.12% | 78.92% | 77.53% | 77.33% |
| $RDVI_{SimCSE}$ w/o $\ell_{E_{text}}$ | 73.06% | 80.31% | 77.76% | 77.84% |
| $RDVI_{SimCSE}$ | 71.37% | **85.39%** | **79.41%** | **79.54%** |

We investigate the impact of different parameters on our models' performance. Table 5 shows the effect of *batch size*. We observe that the performance increases with an increase in batch size, but the improvement plateaus beyond a batch size of 16. Table 6 shows the effect of *learning rate*. We found that the performance did not improve with an increase in learning rate and achieved the best result at a rate of $2 \times e^{-5}$. In Table 7, we examined the effect of selecting different values of *K*. We observe that the detection performance did not increase with the increase in K. This indicates that having too many text segments will not benefit the model but instead increase redundancy and reduce the detection ability of the model. Table 8 presents the effect of *window size*. Empirically, we find that the model achieved the best result when the window size equals 3, while a smaller or larger window size only degrades the model's detection performance.

**Table 5.** The effect of different batch sizes.

| Batch Size | Precision | Recall | F1 Score | Accuracy |
| --- | --- | --- | --- | --- |
| 8 | 70.83% | 81.30% | 77.18% | 77.27% |
| 16 | 71.37% | 85.39% | **79.41%** | **79.54%** |
| 32 | 66.03% | **88.72%** | 78.43% | 78.78% |
| 48 | **73.66%** | 82.13% | 78.72% | 78.78% |

**Table 6.** The effect of different learning rates.

| Learning Rate | Precision | Recall | F1 Score | Accuracy |
| --- | --- | --- | --- | --- |
| $5 \times e^{-6}$ | 60.03% | **88.27%** | 75.50% | 76.10% |
| $2 \times e^{-5}$ | 71.37% | 85.39% | **79.41%** | **79.54%** |
| $5 \times e^{-5}$ | **76.34%** | 78.74% | 77.82% | 77.82% |
| $1 \times e^{-4}$ | 68.70% | 85.31% | 78.19% | 78.39% |

**Table 7.** The effect of different K.

| Top K | Precision | Recall | F1 Score | Accuracy |
| --- | --- | --- | --- | --- |
| 1 | **71.37%** | 85.39% | **79.41%** | **79.54%** |
| 2 | 64.89% | **89.01%** | 78.00% | 78.40% |
| 3 | 70.99% | 84.55% | 78.84% | 78.97% |
| 4 | 70.23% | 83.26% | 77.88% | 78.01% |
| 5 | 67.94% | 85.99% | 78.16% | 78.40% |

**Table 8.** The effect of different window sizes.

| Window Size | Precision | Recall | F1 Score | Accuracy |
| --- | --- | --- | --- | --- |
| 1 | 70.99% | 84.16% | 78.65% | 78.78% |
| 3 | 71.37% | **85.39%** | **79.41%** | **79.54%** |
| 5 | **72.52**% | 84.07% | 79.26% | 79.35% |

To evaluate the ability of our proposed method to identify newly generated sarcasm by retrieving the latest knowledge, we developed a new test set. The data in this set were also crawled from *Guanchazhe* and consist of the most recent news and comments that do not overlap with *GuanSarcasm* datasets. We collected a total of 996 samples from 27 December 2022 to 8 February 2023, which consisted of 357 verbal ironic samples and 599 non-ironic samples. To establish a baseline for comparison, we selected several models that demonstrated good performance on *GuanSarcasm* and directly tested them on the new test set. The results of the experiment are presented in Table 9. Our analysis shows that the performance of all the methods significantly dropped on the new test set, indicating that sarcasm expressions are subject to temporal variations, and models trained on static datasets may not be effective at detecting sarcasm in real-world scenarios. The performance of the chatGPT-based method demonstrates a clear degradation, which can be attributed to the differences between the examples used in in-context learning (ICL) and the test set. This leads to a poor detection performance. The method proposed in this paper demonstrates enhanced performance on the new test set, surpassing existing approaches. One possible explanation is that the expressions and rhetorical techniques employed in sarcasm also evolve over time, which models cannot learn from static datasets.

**Table 9.** Experimental results on new test set.

| Approaches | Precision | Recall | F1 Score | Accuracy |
|:---:|:---:|:---:|:---:|:---:|
| **BERT** | 31.11% | 52.04% | 54.63% | 60.05% |
| **BERT**$^{SSAS}$ | 29.11% | 56.22% | 55.29% | 61.69% |
| **ChatGPT** | **44.67%** | 41.27% | 50.24% | 51.32% |
| **ChatGPT+Retrieval** | 28.67% | 48.13% | 52.43% | 58.14% |
| **RDVI**$_{BM25}$ | 30.67% | **57.02%** | 56.13% | **62.15%** |
| **RDVI**$_{SimCSE}$ | 40.89% | 52.27% | **57.40%** | 60.51% |

To qualitatively demonstrate that our method can retrieve relevant segments as connotative knowledge to improve the model's ability to detect verbal irony, we chose three examples and displayed the corresponding text fragments retrieved by the retriever, as illustrated in Table 10. In the first example, a plane from India crashed in Indian-administered Kashmir, and the given text was: "Falling down and getting up makes one stronger!" The retriever has retrieved relevant information about the Kashmir region, which can help the model better understand the context of the event. In the second example, the retriever provides additional details on the significant military expenditures of the US government in recent years, enabling the model to grasp that the given text expresses discontent with the government's significant military spending. In the third example, the retriever retrieves information on the corruption problem of the US military in Afghanistan, allowing the model to comprehend the meaning of the given text better. These examples illustrate that the retrieved information represents connotative knowledge present in the text, and integrating such knowledge can improve the model's understanding of the given text.

**Table 10.** The Case Study. We selected three different examples to showcase the most relevant text segments found by the retriever, and words in text segments highlighted in red are entities directly related to the given text or its context.

| Index | Given Text | Context | Connotative Knowledge |
|:---:|:---:|:---:|:---:|
| 1 | Falling down and getting up makes one stronger! | An Indian fighter jet crashed in the Indian-controlled Kashmir region. | The region is divided amongst three countries in a territorial dispute: Pakistan controls the northwest portion (Northern Areas and Kashmir), India controls the central and southern portion (Jammu and Kashmir) and Ladakh ... |
| 2 | Americans are having a great time playing the arms race game by themselves. | Cutting Equipment Purchases, the US Department of Defense allocates $100 billion for research and development. | The United States has deployed overseas troops in multiple countries and regions around the world, totaling over 230,000 personnel. Currently, the US is the country with the highest military expenditure in the world, ... |
| 3 | Keep going. I believe in you. | US military officials claimed that the political situation in Afghanistan does not allow for the withdrawal of US troops. | After years of military operations yielding little results, the United States decided to withdraw from Afghanistan in 2014. The new Afghan government supported by the US was plagued by corruption issues... |

Then, we leverage t-SNE [64] to visualize the representation embedding of our model and BERT. Through Figure 2, we find that our model can learn a high-quality representation to facilitate the performance of sarcasm detection.

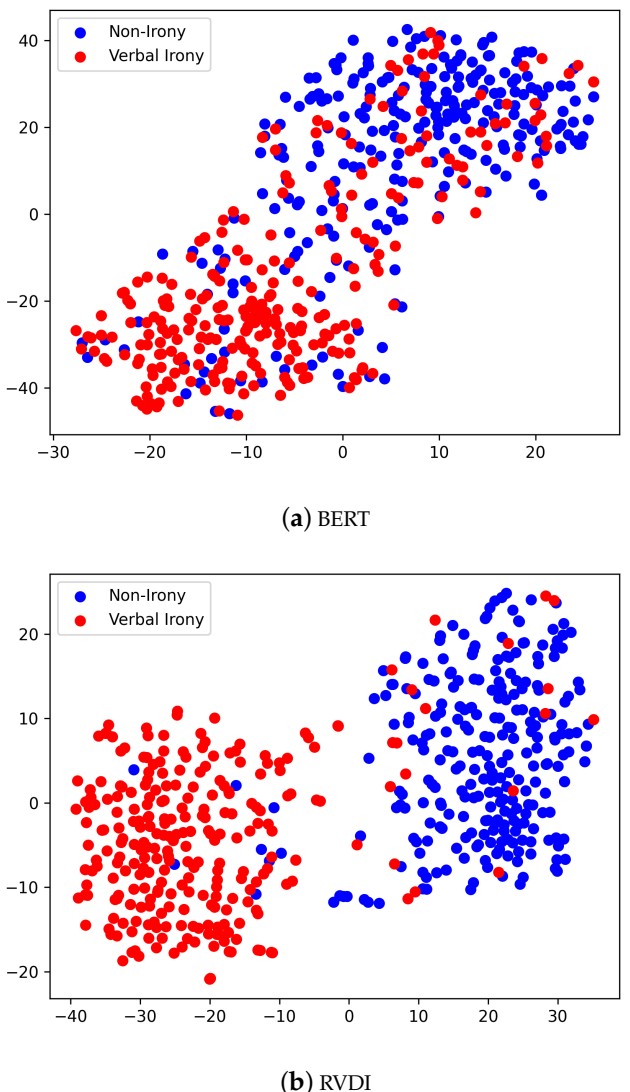

(**a**) BERT

(**b**) RVDI

**Figure 2.** The t-SNE visualization of the representation embeddings.

## 5. Conclusions

In this work, we propose a two-stage retrieval-detection framework, called *RDVI*, that utilizes connotative knowledge to enhance the detection of verbal irony. In the first stage, it retrieves documents with relevant connotative knowledge and selects the most similar segments that could contain connotative knowledge. In the second stage, connotative knowledge is employed through prompt learning to improve the model's semantic comprehension, ultimately enhancing its ability to detect verbal irony. Experimental results demonstrate that our method effectively incorporates connotative knowledge through retrieval and prompt learning to facilitate the capacity of verbal irony detection.

**Author Contributions:** Conceptualization, Z.W. and R.W.; methodology, Z.W.; software, R.W.; validation, S.C. and Q.W.; formal analysis, Q.W. and K.D.; data curation, B.L.; writing—original draft preparation, Z.W.; writing—review and editing, S.C. and K.D.; visualization, B.L.; supervision, R.X. All authors have read and agreed to the published version of the manuscript

**Funding:** This research was funded by the National Natural Science Foundation of China (61876053, 62006062, 62176076), the Guangdong Provincial Key Laboratory of Novel Security Intelligence Technologies (2022B1212010005), the Shenzhen Foundational Research Funding (JCYJ20200109113441941,

JCYJ20210324115614039), Shenzhen Science and Technology Program JSGG20210802154400001, and the Joint Lab of HITSZ and China Merchants Securities.

**Data Availability Statement:** The partition version of the *GuanSarcasm* dataset generated during the current study is available from the corresponding author upon reasonable request.

**Conflicts of Interest:** The authors declare no conflicts of interest.

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
