# Peer review of "RDVI: A Retrieval–Detection Framework for Verbal Irony Detection"

_electronics, doi:10.3390/electronics12122673_

Round 1

Reviewer 1 Report

The authors present "RDVI: A Retrieval-Detection Framework for Verbal Irony Detection" utilizing open domain knowledge and prompt learning to improve irony detection. The study proposes an architecture and reveals comparative results by benchmarking against available solutions. Their proposal performs better than some SOTA models on the same task. 

Below please find my comments:

########################################

Abstract

"Accurately recognizing verbal irony is essential to enhance the understanding of users’ true intentions, thereby facilitating more natural language processing tasks."

-  I suggets rephrasing along the lines...: Accurately recognizing verbal irony is essential for any NLP application for which the understanding of the true user intentions are key to perform the underlying taks. 

"the model's": 

- I suggest using "a model's" and furthermore think about being more specific. Which model type?

"To address this issue, we propose a verbal irony detection method based on a retrieval-detection framework called RDVI."

- Sounds as if your proposal was based on an existing method calles RDVI. I suggest: "..., we propose a Retrieval-Detection method for Verbal Irony (RDVI)....." 

################

Line 26: "including e.g. sentiment analysis[6], hate speech detection [7], and argument detection [8]".

Line 32: Consider changing table order. Is weird to start with Table 10. In addition, consider to add an example directly in the flow of the text at this position as looking it up in the table breaks the reading flow.

Ok, now that I reached line 37 is became obvious that the first table was a typo and thus wrong link. 

In that case, and the table probably being printed close to the reference, the reading flow is not that much affected. Up to you. 

Line 52: "knowledge in the above example is ...". Refer more explicitly to the example (Samsung)

Lines 57-58: Please add references if available

Line 62: "The existing OpenQA system is usually consists of two primary components" - Check grammar

Line 72: Was time, other than in the abstract, that the acronym 'RDVI' is used. Rewrite accordingly. 

Lines 67-73: Avoid one-sentence paragraphs

Lines 81-92: Is this requested by the journal's style? If not, I recommend removing the bullet points and integrate the information seamlessly with the text. Also, remove lines 89-92 as they do not add value to the introduction and I have rarely seen this style. Be aware that if writing it that way is journal style, ignore my comment. 

Line 95: Remove this sentence as it is highly superficial. Start with the second sentence. 

Line 114: "In the subsequent study, " ---- it is unclear to this reviewer why you use "subsequent" as you do not specify one, but reference several studies. 

Line 144: Typo

Line 183: "shows" to "is shown"

Lines 176-190: You have not yet introduced the 'model(s)'. Please introduce the model in an additional paragraph. Also differentiate better between model and architecture in the text. "The overall architecture of our model shows in Figure 1.". I guess the authors want to say that the architecture of their overall approach is presented in Figure 1. The 'model' as this reviewer understands is either the SimCSE or the binary clasifier used in the detection stage (PLM). Also, the abbreviation 'PLM' (pre-trained language model) is not explained anywhere. The figure legend should be a bit more explanatory. 

Line 199: Recognize is uppercase. If intentionally, also italic as you do the paragraph below for Retrieve

Lines 192-252: I really like those explanations. I suggest the authors use the numbering also in the Figure. You could then use the figure legend to point to the more detailed explanations given in this paragraph. 

Lines 254-256 and lines 258-260 do contain almost the same information. Comsider referencing the dataset in 154-256, remove lines 258-259, and start the paragraph with "GuanSarcasm was manually annotated by"...

Line 277: "sequence length is" --> was

Lines 281-302: Consider using a differnt type of structure, the current one is not very appealing. Either remove the line breaks or bullet points?

Line 312: ur model's performance, we conducted the

Lines 313-315: "The results indicate that our method RDVISimCSE is statistically significant at the 0.001 level(t = 11.353, p = 3.267e − 06 < 0.001) for the best baseline BERTSSAS." You are switching in tense. Also consider rephrasing this sentence. This author, from only reading this sentence, does not understand how the t-text was performed. The f1-scores that were you used to test your model against the baseline were obtained how? You only tested on one dataset, which results in one f1-value. Or did you compare f1-scores from different batch sizes, learning rates etc? Please clarify. 

Line 318: When --> when

Line 330-341....: Switching in tense, I suggest to stick to past tense. 

Line 345-348: That is an important oberservation and a typical issue regarding the generalization of many types of NLP applications that require constant adjustment. 

Line 350: I think is the first time appearance of the acronym 'ICL'.

Line 351: "The method proposed in this paper can achieve better performance, but the obtained results require further improvements." Rewrite, not clear to this reviewer what this sentence wants to say. 

Lines 338-348: I would add this information to the Dataset paragraph. Here, it almost comes out of nowhere. 

Figure 2 and 3 are nice visualizations. Please add axis labels and consider merging into one Figure with A) and B).

Lines 372 following......   tense!

In general, consider to have different sections for Results and Discussion, unless journal style prefers this format. 

The English of the presented article is ok. Some easy to fix typos and formatting issues. Editing is needed regarding the use of past vs. present tense to be consistent.

Reviewer 2 Report

The paper is well written, bar few minor language edits. It is also exploring a very critical aspect of language processing, irony, using state of art techniques.

The only reservation I have is the use of the word "Verbal" in the title as well as the rest of the paper. It implies spoken signal, which has many more dimenstions then written text. In fact the data used from Guanchazhe Chinese Sarcasm Dataset is written rather than spoken.

The quality of language is excellent, bar few required edits.

Reviewer 3 Report

The paper proposes a novel Retrieval-Detection Framework for Verbal Irony Detection (RDVI) that combines retrieval-based and detection-based approaches to improve the accuracy of verbal irony detection. 

The paper addresses the importance of detecting verbal irony in natural language processing, particularly in social media contexts where it is prevalent. The authors argue that accurate detection of verbal irony can help improve sentiment analysis, opinion mining, and other NLP tasks. 

The paper is well-organized and clearly presents the proposed framework, related work, experiments, and results. The authors provide a thorough evaluation of their proposed framework using several baseline methods and datasets. However, it would be nice to include the GPT4 model in the experimental results if possible.

The proposed framework offers a new approach to detecting verbal irony that could potentially improve the accuracy of NLP tasks in social media contexts. As a result, the SimCSE method seems to have played a significant role in improving performance. However, it would be good to add a discussion section to explain the limitations of the proposed model and how it compares to other models.

The quality of the English is not a problem for reading this paper. 
